# Kick Bad Guys Out! Conditionally Activated Anomaly Detection in Federated Learning with Zero-Knowledge Proof Verification

## Abstract

Federated Learning (FL) systems are susceptible to adversarial attacks, such as model poisoning attacks and backdoor attacks. Existing defense mechanisms face critical limitations in real-world deployments, such as relying on impractical assumptions (*e.g.*, adversaries acknowledging the presence of attacks before attacking) or undermining accuracy in model training, even in benign scenarios. To address these challenges, we propose REDJASPER, a two-staged anomaly detection method specifically designed for real-world FL deployments. It identifies suspicious activities in the first stage, then activates the second stage conditionally to further scrutinize the suspicious local models, employing the $3\sigma$ rule to identify real malicious local models and filtering them out from FL training. To ensure integrity and transparency within the FL system, REDJASPER integrates zero-knowledge proofs, enabling clients to cryptographically verify the server's detection process without relying on the server's goodwill. REDJASPER operates without unrealistic assumptions and avoids interfering with FL training in attack-free scenarios. It bridges the gap between theoretical advances in FL security and the practical demands of real-world deployment. Experimental results demonstrate that REDJASPER *consistently delivers performance comparable to benign cases*, highlighting its effectiveness in identifying potential attacks and eliminating malicious models with high accuracy.

## 1 Introduction

Federated Learning (FL) (McMahan et al., 2017) is vulnerable to various security threats (Cao & Gong, 2022; Bhagoji et al., 2019; Lam et al., 2021; Jin et al., 2021; Tomsett et al., 2019; Chen et al., 2017; Tolpegin et al., 2020a; Kariyappa et al., 2022; Zhang et al., 2022d). Malicious clients may deliberately manipulate their local models to disrupt global model convergence (Fang et al., 2020; Chen et al., 2017) or implant backdoors that cause the global model to misclassify specific inputs (Bagdasaryan et al., 2020b;a; Wang et al., 2020). These threats undermine the reliability of FL in real-world scenarios, where the participation of adversarial participants may be unpredictable and their malicious intent remains hidden until an attack is successfully executed.

Existing FL defense mechanisms face critical limitations in practical deployment (Blanchard et al., 2017; Yang et al., 2019; Kumari et al., 2023; Fung et al., 2020; Pillutla et al., 2022; He et al., 2022; Cao et al., 2022; Karimireddy et al., 2020; Sun et al., 2019; Fu et al., 2019; Ozdayi et al., 2021; Sun et al., 2021; Yin et al., 2018; Chen et al., 2017; Xie et al., 2020; Li et al., 2020; Cao et al., 2020; Yu et al., 2023; Zhang & Li, 2024; Nguyen et al., 2022; Rieger et al., 2022). These defenses might rely on impractical assumptions or require unrealistic prior knowledge (Blanchard et al., 2017; Sun et al., 2019; Fu et al., 2019; Ozdayi et al., 2021). For example, Krum (Blanchard et al., 2017) assumes that the server is aware of the number of malicious clients and the timing of them attacking, an assumption that rarely holds in practice. Adversaries actively conceal their malicious intent and will not notify the FL system before attacking. Other defenses attempt to improve robustness by modifying local models and/or model aggregation, such as adjusting aggregation functions (Pillutla et al., 2022), re-weighting client updates (Fung et al., 2020; Nguyen et al., 2022; Rieger et al., 2022), or discarding suspicious models (Blanchard et al., 2017). However, these strategies often degrade model performance even in attack-free scenarios. Given that attacks are typically infrequent in

Figure 1: Overview of REDJASPER

real-world settings, interference with aggregation aggressively penalizes honest participants and undermines accuracy across all training round. Furthermore, most defenses operate exclusively on the FL server without providing mechanisms for clients to verify their correct execution (Fung et al., 2020; Nguyen et al., 2022; Rieger et al., 2022; Blanchard et al., 2017). The honest clients have to trust the server blindly, undermining transparency and accountability in FL systems.

To ensure practicality in real-world deployments, FL defenses must satisfy three requirements: *i*) the defense should operate on-demand, activated only when attacks might have happened to avoid interference during benign training rounds; *ii*) upon detecting a potential attack, the defense should accurately identify malicious local models and mitigate or eliminate their negative impact without harming benign ones; and *iii*) the defense should include a verification mechanism that enables clients to validate the integrity of server-side operations without relying solely on the server's trustworthiness.

Table 1: Comparison with state-of-the-art defenses.

| Method | Attack presence detection | Removing malicious models | Free from impractical knowledge | Free from reweighting | Free from aggregation modification | Free from harming benign models | *Robust performance in non-attack scenarios* | Execution Integrity Verification |
|---|---|---|---|---|---|---|---|---|
| **Krum** (Blanchard et al., 2017) | ✗ | ✓ | ✗ | ✓ | ✓ | ✗ | ✗ | ✗ |
| **RFA** (Pillutla et al., 2022) | ✗ | ✗ | ✓ | ✓ | ✗ | ✗ | ✗ | ✗ |
| **Foolsgold** (Fung et al., 2020) | ✗ | ✗ | ✓ | ✗ | ✓ | ✗ | ✗ | ✗ |
| **NormClip** (Sun et al., 2019) | ✗ | ✗ | ✓ | ✓ | ✓ | ✗ | ✗ | ✗ |
| **Bucketing** (Karimireddy et al., 2020) | ✗ | ✗ | ✓ | ✗ | ✓ | ✗ | ✗ | ✗ |
| **Median** (Yin et al., 2018) | ✗ | ✗ | ✓ | ✓ | ✗ | ✓ | ✗ | ✗ |
| **TrimMean** (Yin et al., 2018) | ✗ | ✗ | ✓ | ✓ | ✓ | ✗ | ✗ | ✗ |
| **Flip** (Zhang et al., 2022a) | ✗ | ✗ | ✗ | ✓ | ✓ | ✗ | ✗ | ✗ |
| **Snowball** (Qin et al., 2024) | ✗ | ✓ | ✓ | ✓ | ✗ | ✓ | ✗ | ✗ |
| **Flame** (Nguyen et al., 2022) | ✗ | ✗ | ✓ | ✓ | ✗ | ✗ | ✗ | ✗ |
| **DeepSight** (Rieger et al., 2022) | ✓ | ✓ | ✓ | ✓ | ✗ | ✗ | ✗ | ✗ |
| **Ours** | ✓ | ✓ | ✓ | ✓ | ✓ | ✓ | ✓ | ✓ |

This paper presents REDJASPER, a two-stage defense mechanism designed to detect and filter malicious client models in during each FL training round while addressing the unique challenges of real-world deployment. As illustrated in Figure 1, REDJASPER begins with a ***cross-round detection*** stage that monitors round-to-round behaviors to identify suspicious deviations from normal training dynamics. Upon detection of suspicious activities, REDJASPER activates ***cross-client detection*** that quantifies the maliciousness (*i.e.*, the "*evilness level*") of each local model and filters out the malicious ones based on the $3\sigma$ rule (Pukelsheim, 1994). To ensure transparency and integrity, REDJASPER integrates Zero-Knowledge Proofs (ZKPs) (Goldwasser et al., 1989) to provide cryptographic guarantees of its honest execution on the FL server. We compare REDJASPER against a wide range of state-of-the-art defenses (Blanchard et al., 2017; Pillutla et al., 2022; Fung et al., 2020; Sun et al., 2019; Karimireddy et al., 2020; Yin et al., 2018; Zhang et al., 2022a; Qin et al., 2024) in Table 1. Our key contributions are as follows:

*i*) **Practical Applicability in Real-World FL Systems:** REDJASPER operates without requiring impractical prior knowledge (*e.g.*, the number of attackers or the timing of attacks) and is explicitly designed for real-world deployment. To the best of our knowledge, this is the first method that explicitly bridges the gap between academic research and real-world applications of FL security.

*ii*) **Conditional Activation:** REDJASPER does not interfere with training in attack-free scenarios, a critical requirement for real-world deployments where adversarial behavior is infrequent and model accuracy is paramount. It activates detection and filtering only when attacks are suspected, avoiding disturbing benign training and preventing accuracy degradation.

*iii*) **Non-Disruptive Operation:** Upon identification of suspicious activities, REDJASPER detects and removes malicious local models with high accuracy, without modifying the aggregation function or negatively impacting benign local models.

*iv*) **Enhanced Detection Accuracy:** REDJASPER avoids removing local models based on scores directly computed with models. Instead, it leverages the statistical properties of these scores and applies the $3\sigma$ rule to precisely identify malicious local models, thereby enhancing detection accuracy and preserving benign submissions.

*v*) **Verifiability:** REDJASPER enables clients to independently verify the integrity of server-side defense operations with ZKP, fostering accountability without relying on the server's goodwill.

## 2 PROBLEM SETTING

### 2.1 ADVERSARY MODEL

We consider an FL system in which a subset of participating clients may be adversarial and attempt to tcompromise the training process to achieve some malicious goals. Adversarial behaviors include: *i*) injecting backdoors into local updates to cause the global model to misclassify specific inputs (Bagdasaryan et al., 2020b; Wang et al., 2020; Yu et al., 2023); *ii*) performing Byzantine attacks by intentionally manipulating local models to prevent the global model from converging (Chen et al., 2017; Fang et al., 2020); and *iii*) submitting fabricated or untrained models to the server (Wang, 2022). We further assume that the adversaries might be *adaptive*, *i.e.*, they can observe the defense mechanism and adapt their attack strategies accordingly (Wu et al., 2023). Following common threat model assumptions in literature (Blanchard et al., 2017; Ozdayi et al., 2021; Sun et al., 2021; Yin et al., 2018; Chen et al., 2017; Xie et al., 2020), we assume at least 50% clients are benign, and assume the FL server is not fully trustworthy, consistent with the realistic deployment scenarios with potentially untrusted execution environments. While clients expect the server to deploy a defense mechanism, they remain uncertain about whether it is executed faithfully.

### 2.2 PRELIMINARIES

**Federated Learning (FL).** FL (McMahan et al., 2017) enables training machine learning models across decentralized devices without centralizing their raw data. FL is particularly beneficial when dealing with sensitive data, as it allows data to remain on the client device during training.

**Krum.** Krum or its variant $m$-Krum (Blanchard et al., 2017) selects one (or $m$) local model(s) whose updates are closest to the majority, based on pairwise distances, for aggregation. See Appendix §A for further details.

**$3\sigma$ Rule.** $3\sigma$ (Pukelsheim, 1994) is an empirical rule commonly used in anomaly detection for data management tasks (Han et al., 2019). It states that approximately 68%, 95%, and 99.7% of data values lie within one, two, and three standard deviations from the mean, respectively, under a normal distribution. This rule is broadly applicable in real-world settings, as many real-world data distributions approximate normality (Lyon, 2014). Moreover, even when the data is not normally distributed, transformation techniques can be applied to approximate a normal distribution (Aoki, 1950; Osborne, 2010; Sakia, 1992; Weisberg, 2001).

**Zero-Knowledge Proofs (ZKPs).** A ZKP (Goldwasser et al., 1989) is a proof system enabling a prover to convince a verifier that a function has been correctly computed on the prover's secret input (witness). ZKPs have three properties: *i*) *correctness*: the proof they produce should pass verification if the prover is honest; *ii*) *soundness*: a cheating prover cannot convince the verifier with overwhelming probability, and *iii*) *zero-knowledge*: the prover's witness is not learned by the verifier.

## 3 REDJASPER: A TWO-STAGE ANOMALY DETECTION MECHANISM

REDJASPER operates in each FL round after the server collects local models. It first performs a lightweight *cross-round check* to assess the likelihood of potential attacks, then, if suspicious activity is detected, activates a more rigorous *cross-client detection* phase that evaluates the maliciousness,

*i.e.*, the *evilness level*, of each local model. Malicious models are then removed using the $3\sigma$ rule to mitigate their impact on the global model. Below we describe the two phases in detail.

---

**Algorithm 1** REDJASPER-Phase 1: Cross-Round Detection

---

**Input:** $\tau$: training round ID ($\tau = 0, 1, 2, \ldots$); $\mathcal{W}^\tau$: client models of round $\tau$; $\gamma$: similarity threshold.

1: **if** $\tau = 0$ **then return True**       ▷ *No previous round, activate cross-client detection*
2: $\mathcal{W}^{\tau-1} \leftarrow$ get_cached_client_models(), $\mathbf{w}_g^{ref} \leftarrow$ get_global_model_of_last_round()
3: **for all** $\mathbf{w}_i^\tau \in \mathcal{W}^\tau$ **do**
4:      $\mathcal{S}_c(\mathbf{w}_i^{\tau-1}, \mathbf{w}_i^\tau) \leftarrow$ get_similarity($\mathbf{w}_i^{\tau-1}, \mathbf{w}_i^\tau$), $\mathcal{S}_c(\mathbf{w}_g^{ref}, \mathbf{w}_i^\tau) \leftarrow$ get_similarity($\mathbf{w}_g^{ref}, \mathbf{w}_i^\tau$)
5:      **if** $\mathcal{S}_c(\mathbf{w}_g^{ref}, \mathbf{w}_i^\tau) < \gamma$ **or** $\mathcal{S}_c(\mathbf{w}_i^{\tau-1}, \mathbf{w}_i^\tau) < \gamma$ **then return True**       ▷ *Potential attacks*
6: **return False**       ▷ *No suspicious attacks*

---

### 3.1 CROSS-ROUND DETECTION

Cross-round detection serves as a "gatekeeper" that evaluates the likelihood of suspicious activities in local models, such that REDJASPER can decide whether to activate the next phase for more rigorous detection and potential removal of malicious models.

The cross-round detection computes cosine similarities between the local models of the current round and some reference models. Two types of reference models are involved, including *i*) the global model of the last FL training round; and *ii*) verified benign local models identified from the previous rounds. These models have a high likelihood of being benign thus can serve as a reliable *golden truth* for the cross-round check. The global model provides a reference for convergence; local models that deviate significantly from the expected global model may be attempting to disrupt training. Meanwhile, comparing clients' current submissions with their previously verified benign submissions allows the detection of sudden behavioral shifts, *e.g.*, transitioning from benign to malicious behaviors, thereby flagging inconsistencies in client-specific updates across consecutive FL rounds.

We illustrate the idea in Figure 2. Benign local models are expected to exhibit high similarities to the reference models. For each local model $\mathbf{w}_i$ and a reference model $\mathbf{w}_r$, the cosine similarity is computed as $\mathcal{S}_c(\mathbf{w}_i, \mathbf{w}_r) = \frac{\mathbf{w}_i \cdot \mathbf{w}_r}{||\mathbf{w}_i|| \cdot ||\mathbf{w}_r||}$. A high similarity reflects strong alignment between $\mathbf{w}_i$ and $\mathbf{w}_r$, indicating the client is more likely to be benign. Lower similarities, on the other hand, signal potential adversarial behaviors, as malicious clients might submit

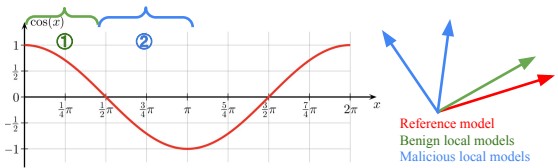

Figure 2: Cosine similarities. ① indicates likely benign models with high cosine similarity, and ② indicates likely malicious models with low cosine similarity.

manipulated models that diverge $\mathbf{w}_i$ from $\mathbf{w}_r$ (Chen et al., 2017; Fang et al., 2020; Bagdasaryan et al., 2020b; Wang et al., 2020). In practice, REDJASPER employs a threshold $\gamma$ ($-1 < \gamma < 1$). Similarity scores lower than $\gamma$ indicate potential adversarial behaviors of the corresponding clients and will activate a further inspection in the second phase, as described later in §3.2.

The cross-round detection algorithm is summarized in Algorithm 1. Upon the server receiving local models from clients, it first loads the reference models, including the client models of the last round and the global model of the last round. Then, it computes cosine similarities between each local model and the corresponding reference. Local models exhibiting higher similarities to these references are deemed more likely to be benign, while those with lower similarities are considered suspicious, activating a rigorous cross-client detection in the next phase. We note that REDJASPER just flags suspicious models without removing them, thus, REDJASPER does not rely heavily on cosine similarities.

We leverage a modified Positive Predictive Value (PPV) (Fletcher, 2019) to evaluate the accuracy of cross-round detection while revealing whether all malicious models are identified.

**Definition 3.1.** Let $\mathcal{T}$ be an FL training process consisting of $\tau$ rounds ($\tau > 0$). Denote by $\mathcal{W}$ the set of all client submissions across all rounds, partitioned into malicious submissions $\mathcal{W}_{\text{bad}}$ and benign submissions $\mathcal{W}_{\text{good}}$. Let $\mathcal{W}_{\text{bad}}^d$ and $\mathcal{W}_{\text{good}}^d$ represent the sets of submissions detected as *malicious*

and *benign*, respectively, by a detection mechanism $\mathcal{M}$. Then true-positives (TP) is defined as $N_{TP} = \left| \mathcal{W}^d_{\text{bad}} \cap \mathcal{W}_{\text{bad}} \right|$, and false-positives (FP) is defined as $N_{\text{FP}} = \left| \mathcal{W}^d_{\text{bad}} \cap \mathcal{W}_{\text{good}} \right|$. We define a modified PPV as $\text{PPV} = \frac{N_{\text{TP}}}{N_{\text{TP}} + N_{\text{FP}} + |\mathcal{W}_{bad}|}$, where $0 \leq \text{PPV} \leq \frac{1}{2}$.

---

**Algorithm 2** REDJASPER-Phase 2: Cross-Client Detection

---

**Input:** $\tau$: training round ID ($\tau = 0, 1, \ldots$); $\mathcal{W}^\tau$: local models of round $\tau$; $m$: parameter of $m$-Krum; $\lambda$: parameter of $3\sigma$ Rule; $\mathbf{w}^{ref}_g$: global reference model from the previous round.

1: **if** $\tau = 0$ **then** $m \leftarrow |\mathcal{W}^\tau|/2$, $f \leftarrow |\mathcal{W}^\tau|/2$, $\mathbf{w}^{ref}_g \leftarrow \text{Krum\_and\_m\_Krum}(\mathcal{W}^\tau, m, f)$

2: $\mathcal{L} \leftarrow \text{compute\_L2\_scores}(\mathcal{W}^\tau, \mathbf{w}^{ref}_g)$, $\mu \leftarrow \frac{\sum_{\ell \in \mathcal{L}} \ell}{|\mathcal{L}|}$, $\sigma \leftarrow \sqrt{\frac{\sum_{\ell \in \mathcal{L}} (\ell - \mu)^2}{|\mathcal{L}| - 1}}$ ▷ *Estimate $\mathcal{N}(\mu, \sigma)$*

3: **for all** $\mathbf{w}_i \in \mathcal{W}^\tau$ **do**

4:     **if** $\mathcal{L}[i] > \mu + \lambda\sigma$ **then** Remove $\mathbf{w}_i$ from $\mathcal{W}^\tau$

5: Renew $\mathbf{w}^{ref}_g$ for the next round

6: **return** $\mathcal{W}^\tau$                             ▷ *Cache and return the filtered set*

---

Ideally, PPV is $\frac{1}{2}$, indicating perfect performance, *i.e.*, all malicious models are identified ($N_{\text{TP}} = |\mathcal{W}_{\text{bad}}|$) and no benign models are misclassified ($N_{\text{FP}} = 0$). See Appendix §B for details.

### 3.2 CROSS-CLIENT DETECTION

Cross-client detection is activated only if the cross-round detection phase flags potential threats, aiming to verify whether actual attacks have occurred in the current FL round. This phase computes an $L_2$ distance-based *evilness level* of each local model, measuring its deviation from reference behavior. The $3\sigma$ rule is then applied to these scores to identify outliers, which are treated as malicious models and excluded from aggregation.

The cross-client detection is detailed in Algorithm 2. For each local model $\mathbf{w}^\tau_i$ submitted in the current round $\tau$, we compute its *evilness level* with the $L_2$ distance as $||\mathbf{w}^\tau_i - \mathbf{w}^{\tau-1}_g||_2$, where $\mathbf{w}^{\tau-1}_g$ denotes the aggregated global model from round $\tau - 1$. Since the first round lacks a previously aggregated global model, we employ $m$-Krum (Blanchard et al., 2017) to prevent the influence of any potentially malicious models. Specifically, we select half of the local models to compute an approximate average model for initialization. In later training rounds, we do not need $m$-Krum as we can simply utilize the average model from the previous round. With the $L_2$ scores, the algorithm then estimates a normal distribution and applies the $3\sigma$ rule to filter out local models. Since models with lower *evilness levels* are preferable, we apply a one-sided threshold: only models with scores higher than $\mu + \lambda\sigma$ ($\lambda > 0$) are removed, while models with scores lower than the other side of the bound ($\mu - \lambda\sigma$) are retained. The following theorem states that the likelihood of mistakenly identifying a benign client as malicious decreases exponentially with $\lambda$. We defer its proof to Appendix §C, and defer the discussion on why $3\sigma$ rule is effective in identifying malicious models to Appendix §D.

**Theorem 3.2.** *Let $\mathcal{L}$ be the* evilness level *scores for client models in the current FL round, where $\mathcal{L}$ follows normal distribution $\mathcal{N}(\mu, \sigma)$. The* evilness level *for each client $i$ is computed as $\mathcal{L}[i] = ||\mathbf{w}^\tau_i - \mathbf{w}^{\tau-1}_g||_2$. Under CLT, the probability that a benign client is erroneously flagged as malicious using the threshold $\mu + \lambda\sigma$ is upper bounded as $P(\mathcal{L}[i] > \mu + \lambda\sigma) \leq \frac{1}{\sqrt{2\pi}\lambda} e^{-\lambda^2/2}$.*

**Optimization with importance layers.** To perform the detection efficiently, in both the cross-round detection and the cross-client detection, REDJASPER relies on *importance layers* (Fung et al., 2020) of models, *i.e.*, segmental representations of models rather than full model parameters, as references of full models in computation. Specifically, it employs the second-to-last layer, as it retains substantial model information. We detail the impotance layers in Appendix §E and experimentally validate its effectiveness in **Exp 1** in §5.

**Extensions against adaptive attacks.** To extend REDJASPER to be robust against adaptive attacks, cached global models from previous FL rounds cannot be used, as the global model is distributed to all clients and can be exploit by malicious participants. To address this issue, we adapt REDJASPER to operate without relying on cached models. Details are provided in Appendix §F, with Algorithm 4 addressing cross-round detection and Algorithm 5 addressing cross-client detection.

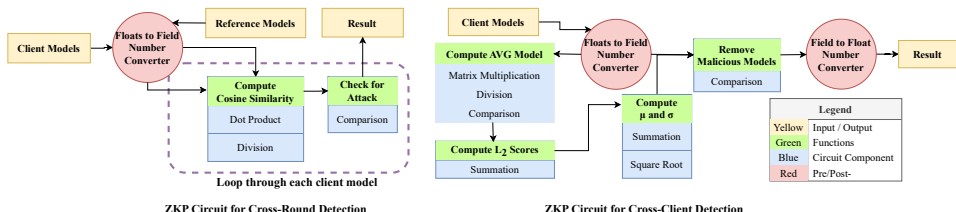

Figure 3: ZKP circuits designed for REDJASPER.

## 4 VERIFIABLE ANOMALY DETECTION

In FL systems with anomaly detection, a critical trust gap arises as clients cannot independently verify the server's execution of the defense mechanism, forcing them to rely on the server's integrity without verifiable assurance. To address this challenge, we integrate ZKPs that enable a prover (*i.e.*, the FL server) to demonstrate computational correctness to verifiers (*i.e.*, clients) without revealing sensitive inputs, such as individual client models or detection thresholds. ZKPs bridge the trust gap in the FL systems with two critical properties: *i*) *Client-Side Verification:* clients can independently verify that the anomaly detection was executed faithfully and in compliance with predefined rules, eliminating reliance on the server's goodwill; and *ii*) *Privacy Preservation:* verification does not require exposing private data, such as other clients' local models or internal server parameters, ensuring confidentiality while maintaining system integrity. We design ZKP circuits as in Figure 3. Our key optimizations are summarized as follows:

*i*) **Freivalds' algorithm Freivalds (1977):** To efficiently verify matrix multiplications (*e.g.*, $AB = C$), we reduce circuit complexity from $O(n^3)$ to $O(n^2)$ by probabilistically checking $A(Bv) \overset{?}{=} Cv$ with a random vector $v$ (Freivalds, 1977; Weng et al., 2021). This method ensures scalable verification without exhaustive recomputation.

*ii*) **Approximate square roots:** To verify that $x = \sqrt{y}$ is computed correctly, we check if $x^2$ is close to $y$ by checking that $x^2 \leq y$ and $(x + 1)^2 \geq y$. This approach reduces the computation of square root to 2 multiplications and 2 comparisons.

The zero-knowledge property ensures public verifiability without exposing sensitive data. ZKPs reveal only the validity of computations, thus, even malicious clients cannot extract sensitive information from proofs. Details of ZKP implementations and motivations are in Appendix §G and Appendix §H, respectively.

## 5 EVALUATIONS

**Models and Datasets.** We evaluate our approach across a diverse set of model–dataset pairs that are widely used in federated learning research. For CV tasks, we adopt CNN (McMahan et al., 2017) on the FEMNIST dataset (Caldas et al., 2018), ResNet-20 (He et al., 2016) on CIFAR-10 (Krizhevsky et al., 2009), and ResNet-56 (He et al., 2016) on CIFAR-100 (Krizhevsky et al., 2009). For NLP tasks, we employ RNN (McMahan et al., 2017) on the Shakespeare dataset (McMahan et al., 2017). Additionally, we also include a lightweight case, *i.e.*, LR (Cox, 1958) on MNIST (Deng, 2012) for real-world application evaluation.

**Setting.** By default, we employ CNN and the non-i.i.d. FEMNIST dataset ($\alpha = 0.5$), as the non-i.i.d. setting closely captures real-world scenarios. We utilize FedAVG in our experiments. We vary the number of clients from 10 to 100 in **Exp 5**, and by default, we use 10 clients for FL training, corresponding to real-world FL applications where the number of clients is typically less than 10, especially in ToB scenarios. For evaluations on adaptive attacks, we leverage ResNet50 and CIFAR100, and set the proportion of malicious clients to 40% by default. We implement the ZKP system in Circom (Contributors, 2022). We conduct our evaluations on a server with 8 NVIDIA A100-SXM4-80GB GPUs, and validate the correct execution with ZKP on Amazon AWS with an m5a.4xlarge instance with 16 CPU cores and 32 GB memory.

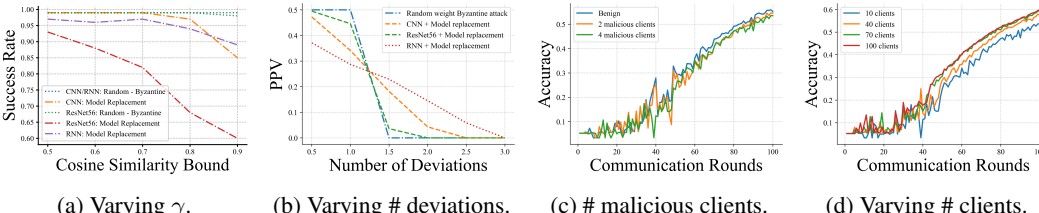

(a) Varying $\gamma$.      (b) Varying # deviations.      (c) # malicious clients.      (d) Varying # clients.

Figure 4: Impacts of different parameters.

**Selection of attacks and defenses.** We employ two Byzantine attacks and two backdoor attacks that are widely considered in the literature, including *i*) a random weight Byzantine attack that randomly modifies the local submissions (Chen et al., 2017; Fang et al., 2020), *ii*) a zero weight Byzantine attack that sets all model weights to zero (Chen et al., 2017; Fang et al., 2020), *iii*) a label flipping backdoor attack that flips labels in the local data Tolpegin et al. (2020b), and *iv*) a model replacement backdoor attack (Bagdasaryan et al., 2020b) that intends to use a poisoned local model to replace the global model. We utilize 5 baseline defense mechanisms that can be effective in real systems, including $m$-Krum (Blanchard et al., 2017), Foolsgold (Fung et al., 2020), RFA (Pillutla et al., 2022), Bucketing (Karimireddy et al., 2020), and Trimmed Mean (Yin et al., 2018). For $m$-Krum, by default, we set $m$ to 5, which means 5 out of 10 submitted local models participate in aggregation in each FL training round. We test our method from the earliest stages of training (*i.e.*, training from scratch), instead of after model convergence, to reflect real-world FL scenarios where adversaries may attack at any point, including during initial model convergence. We do so because early-stage attacks are more challenging: benign local models can exhibit significant variability due to non-i.i.d. data distributions and random initialization. Such variability makes it inherently harder to distinguish malicious models from benign ones, creating a more rigorous testbed for defenses.

**Exp 1: Selection of importance layer.** We utilize the $L_2$ norm of the local models to evaluate the "sensitivity" of each layer. A layer with a norm higher than most of the other layers indicates higher sensitivity compared to others, thus can be utilized to represent the whole model. The results for RNN, CNN, and ResNet-56 are deferred to Figure 11a, Figure 11b, and Figure 11c in Appendix §J, respectively. The results show the sensitivity of the second-to-the-last layer is higher than most of the other layers. Thus, this layer includes adequate information of the whole model and can be selected as the importance layer.

**Exp 2: Impact of the similarity threshold.** We evaluate the impact of the similarity threshold $\gamma$ in the cross-round check with 10 clients in each FL round, where 4 of them are malicious. Ideally, the cross-round check should confirm the absence or presence of an attack accurately. We evaluate the impact of the cosine similarity threshold $\gamma$ in the cross-round check by setting $\gamma$ to 0.5, 0.6, 0.7, 0.8, and 0.9. As described in Figure 4a, the cross-round detection success rate is close to 100% in the case of Byzantine attacks. We observe that, when the cosine similarity threshold $\gamma$ is set to 0.5, the performance is satisfactory in all cases, with at least 93% cross-round detection success rate.

**Exp 3: Selection of the number of deviations** ($\lambda$). We vary $\lambda$ to 0.5, 1, 1.5, 2, 2.5, and 3, and utilize PPV to evaluate the impact of the number of deviations, *i.e.*, the parameter $\lambda$ in the anomaly bound $\mu + \lambda\sigma$. To evaluate a challenging case where a large portion of the clients are malicious, we set 40% clients as malicious in each FL round. Given that the number of FL rounds is 100, the total number of malicious submissions is 400. We evaluate our approach on three tasks, as follows: *i*) CNN+FEMNIST, *ii*) ResNet-56+Cifar100, and *iii*) RNN + Shakespeare. We observe in Figure 4b, that when $\lambda$ is 0.5, the results are the best. Especially for the random weight Byzantine attack, we see that the PPV is exactly 0.5, indicating that all malicious local models are detected. In subsequent experiments, unless specified otherwise, we set $\lambda$ to 0.5.

**Exp 4: Varying the percentage of malicious clients.** We use random Byzantine attack and set the percentage of malicious clients to 20% and 40%. We also include a baseline case where all clients are benign. As shown in Figure 4c, the test accuracy remains relatively consistent across different cases, as in each FL training round, our approach filters out the local models that tend to be malicious to minimize the negative impacts of malicious client models on aggregation.

**Exp 5: Varying the number of clients.** We explore the impact of the number of clients under the random Byzantine attack. We set the number of clients to 10, 40, 70, and 100, and set the percentage

of malicious clients to 40%. The results, as described in Figure 4d, indicate that in all cases, our approach has high utility and can filter out malicious clients with high accuracy.

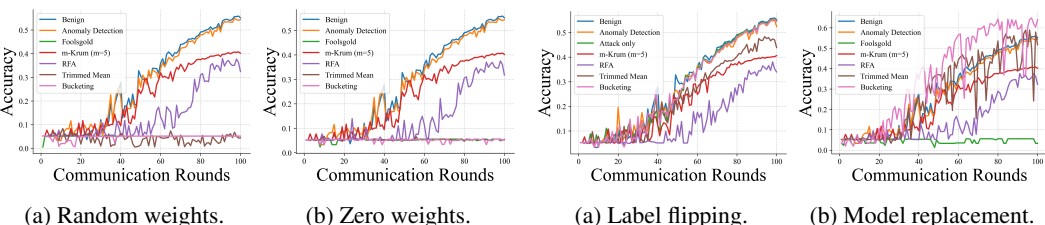

| (a) Random weights. | (b) Zero weights. | (a) Label flipping. | (b) Model replacement. |

Figure 5: Byzantine attacks.          Figure 6: Backdoor attacks.

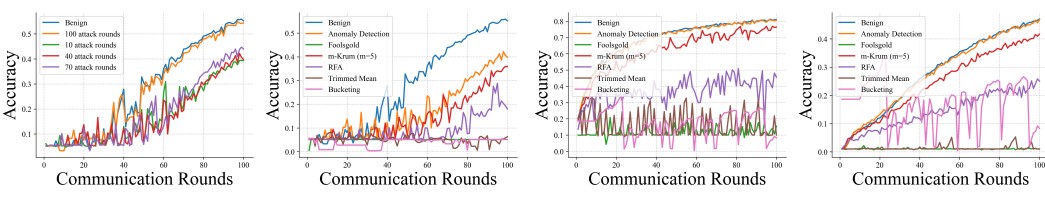

(a) Varying # attack rounds.  (b) 40 attack rounds.  (a) ResNet-20, CIFAR-10. (b) ResNet-56, CIFAR-100.

Figure 7: Evaluations on selected attacks.          Figure 8: Evaluations on CV tasks.

**Exp 6: Evaluations on Byzantine attacks.** We compare our approach with the state-of-the-art defenses and set 10% of the clients as malicious. We include a "benign" case with no activated attack or defense as a baseline. The results for the random weight Byzantine attack (Figure 5a) and the zero weight Byzantine attack (Figure 5b) demonstrate that our approach (shown in orange color) effectively mitigates the negative impact of the attacks and significantly outperforms the other defenses, by achieving a test accuracy much closer to the benign case.

**Exp 7: Evaluations on backdoor attacks.** We compare our approach with the state-of-the-art defenses and set 10% of the clients as malicious. Considering that the label flipping attack is subtle as it manipulates local training data and produces malicious local models that are challenging to detect, we set the parameter $\lambda$ to 2 to produce a tighter boundary. The results for the label flipping attack and model replacement backdoor attack are shown in Figure 6a and Figure 6b, respectively. Results show that our approach is effective against backdoor attacks, with the test accuracy much closer to the benign case compared to the baseline defenses.

**Exp 8: Evaluations on different attack frequencies.** We configure attacks to occur only during specific rounds to evaluate the effectiveness of the proposed two-phase approach. The total number of attack rounds is set to 10, 40, 70, and 100, respectively. We then fix the number of attack rounds to 40 and compare our approach with the state-of-the-art defenses. The results in Figure 7a and Figure 7b show that our method effectively mitigates the impact of the adversarial attacks, ensuring minimal accuracy loss and robust performance even under different attack rounds.

**Exp 9: Evaluations on different tasks.** We evaluate the defenses against the random mode of the Byzantine attack with different models and datasets. The results in Figure 8a, Figure 8b, and Figure 11d in Appendix §J show that our approach outperforms the baseline defenses by effectively filtering out poisoned local models, with a test accuracy close to the benign scenarios. Moreover, some defenses may fail in some tasks, *e.g.*, $m$-Krum fails in RNN in Figure 11d, as those methods either select a fixed number of local models or re-weight the local models in aggregation, which potentially eliminates some local models that are important to the aggregation, leading to an unchanged test accuracy in later FL rounds.

**Exp 10: Evaluations against adaptive attacks with different number of clients.** We evaluate our approach with 10 and 30 clients and compare it to the other defenses, as shown in Figure 9a and Figure 9b. In both scenarios, our method achieves high test accuracy that is close to the benign case and consistently outperforms other approaches, demonstrating its strong robustness against adaptive attacks regardless of the number of clients.

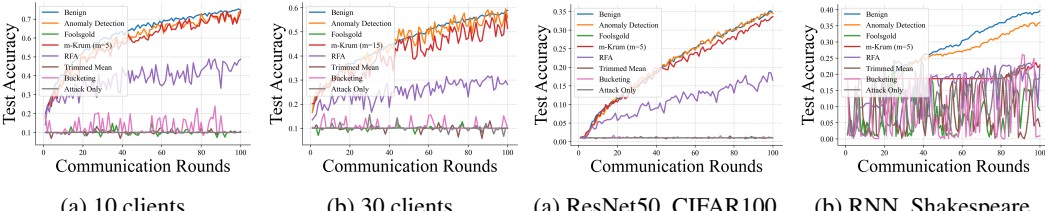

| (a) 10 clients. | (b) 30 clients. | (a) ResNet50, CIFAR100. | (b) RNN, Shakespeare. |

Figure 9: Performance under different # of clients.    Figure 10: Performance under different tasks.

**Exp 11: Evaluations against adaptive attacks across different tasks.** We compare our approach with other defenses on the following tasks: *i*) ResNet50 with CIFAR100; and *ii*) RNN with the Shakespeare dataset; See Figure 10a and Figure 10b. The results show that our approach consistently outperforms other defenses across different tasks, achieving test accuracy close to the benign case. This highlights the effectiveness and generalizability of our method across different tasks.

Evaluations of adaptive attacks under different attacking frequency (**Exp 12**), ZKP verification (**Exp 13**), and evaluations in a real-world setting (**Exp 14**) are deferred to Appendix §J.

## 6 RELATED WORKS

Robust learning and the mitigation of adversarial behaviors in FL has been extensively explored (Blanchard et al., 2017; Yang et al., 2019; Fung et al., 2020; Pillutla et al., 2022; He et al., 2022; Karimireddy et al., 2020; Sun et al., 2019; Fu et al., 2019; Ozdayi et al., 2021; Sun et al., 2021; Yin et al., 2018; Chen et al., 2017; Guerraoui et al., 2018; Xie et al., 2020; Li et al., 2020; Cao et al., 2020). Some approaches keep several local models that are more likely to be benign in each FL round, *e.g.*, (Blanchard et al., 2017; Guerraoui et al., 2018; Yin et al., 2018), and (Xie et al., 2020), instead of aggregating all client submissions. Such approaches are effective, but they keep fewer local models than the real number of benign local models to ensure that all malicious local models are filtered out, causing misrepresentation of some benign local models in the aggregation. This completely wastes the computation resources of the benign clients that are incorrectly removed and thus, changes aggregation results. Some approaches re-weight or modify local models to mitigate the impacts of potential malicious submissions (Fung et al., 2020; Karimireddy et al., 2020; Sun et al., 2019; Fu et al., 2019; Ozdayi et al., 2021; Sun et al., 2021), while other approaches alter the aggregation function or directly modify the aggregation results (Pillutla et al., 2022; Karimireddy et al., 2020; Yin et al., 2018; Chen et al., 2017). Some approaches detect presence of attacks Zhang et al. (2022c) but requires a number of pre-training rounds and relies heavily on historical client models of previous rounds, making it ineffective when there is limited information on past client models. Moreover, the effectiveness of such approach on early rounds of FL training is challenging, as it might require to set several starting round before detection (Zhang et al., 2022b). However, in practice, attacks might happen in early stages of FL training as well. While these defense mechanisms might require unrealistic assumptions or degrade the quality of outcomes due to modifying FL aggregation even in benign cases, thus are not suitable in practical scenarios.

## 7 CONCLUSION

This paper introduces REDJASPER, a verifiable anomaly detection method specifically designed for real-world FL systems. Our method introduces an early cross-round detection step that conditionally activates further anomaly analysis only when attacks are suspected, thereby minimizing unnecessary interference with benign training. REDJASPER enhances the reliability of FL systems, fostering trust among FL participants while promoting positive societal impact. However, it has certain limitations, *e.g.*, it does not support asynchronous FL or vertical FL. Further, while using importance layers can improve efficiency, the proof generation time for ZKPs remains a limitation, restricting wider deployment of this approach and thereby reducing its potential societal benefits. This limitation arises primarily from the inherent limitations of ZKPs. Future advancements in ZKP optimization and hardware acceleration are expected to address this issue.

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

## A  DETAILS OF KRUM AND $m$-KRUM

In Krum and $m$-Krum, the server selects $m$ ($m$ is one in Krum) local models that deviate less from the majority based on their pairwise distances, where such local models are more likely to be benign and thus are accepted for aggregation in the current round. Given that there are $f$ byzantine clients among $L$ clients that participate in each FL iteration, Krum selects one model that is the most likely to be benign as the global model. That is, instead of using all $L$ local models in aggregation, the server selects a single model to represent all $L$ submissions. To do so, Krum computes a score for each model $\mathbf{w}_i$, denoted as $\mathcal{S}_K(\mathbf{w}_i)$, using $L - f - 2$ local models that are "closest" to $\mathbf{w}_i$, and selects the local model with the minimum score to represent the aggregation result. For each local model $\mathbf{w}_i$, suppose $C_i^{\mathcal{N}}$ is the set of the $L - f - 2$ local models that are closest to $\mathbf{w}_i$, then $\mathcal{S}_K(\mathbf{w}_i)$ is computed by

$$\mathcal{S}_K(\mathbf{w}_i) = \sum_{j \in \mathcal{C}_i} ||\mathbf{w}_i - \mathbf{w}_j||^2.$$

An optimization of Krum is $m$-Krum (Blanchard et al., 2017) that selects $m$ local models, instead of one, when aggregating local models. The algorithm for Krum and $m$-Krum is summarized in Algorithm 3 .

---

**Algorithm 3** Krum and $m$-Krum

---

**Input:** $\mathcal{W}$: client submissions of a training round; $m$: number of neighbors considered for computing the Krum score ($m = 1$ for standard Krum); $f$: number of malicious clients in each round.
**Output:** Aggregated global model

1: $\mathcal{S}_k \leftarrow [\,]$      ▷ *List of Krum scores*
2: **for all** $\mathbf{w}_i \in \mathcal{W}$ **do**
3:    $\mathcal{S}_k(\mathbf{w}_i) \leftarrow$ compute_krum_score$(\mathcal{W}, i, m, f)$
4: $\mathcal{W} \leftarrow$ FILTER$(\mathcal{W}, \mathcal{S}_k)$      ▷ *Keep $|\mathcal{W}|/2$ models with lowest scores*
5: **return** average$(\mathcal{W})$
6: **function** COMPUTE_KRUM_SCORE$(\mathcal{W}, i, m, f)$
7:    $d \leftarrow [\,]$      ▷ *List of squared distances*
8:    $L \leftarrow |\mathcal{W}|$      ▷ *Total number of clients*
9:    **for all** $\mathbf{w}_j \in \mathcal{W}$ **do**
10:      **if** $i \neq j$ **then** $d$.append $\left(||\mathbf{w}_i - \mathbf{w}_j||^2\right)$
11:    Sort$(d)$      ▷ *Ascending order*
12:    $\mathcal{S}_k(\mathbf{w}_i) \leftarrow \sum_{k=0}^{L-f-3} d[k]$      ▷ *Use smallest $L - f - 2$ distances*
13:    **return** $\mathcal{S}_k(\mathbf{w}_i)$

---

## B  PROOF OF THE RANGE OF PPV

*Proof.* We leverage a modified PPV evaluate the accuracy of the cross-round detection in identifying potential attacks across all FL training rounds. Below, we show that the upper bound of the modified PPV is $\frac{1}{2}$. We have PPV $= \frac{N_{TP}}{N_{TP} + N_{FP} + |\mathcal{W}_{bad}|}$, thus we have $\frac{1}{PPV} = 1 + \frac{N_{FP}}{N_{TP}} + \frac{|\mathcal{W}_{bad}|}{N_{TP}}$. As $\frac{N_{FP}}{N_{TP}} \geq 0$ and $\frac{|\mathcal{W}_{bad}|}{N_{TP}} \geq 1$, we have $\frac{1}{PPV} \geq 2$, thus PPV $\leq \frac{1}{2}$.      □

## C  PROOF OF THEOREM 3.2

*Proof.* Let $\mathbf{w}_i \in \mathbb{R}^d$ be the local model parameters of client $i$, and $\mathbf{w}_g = \frac{1}{n} \sum_{j=1}^n \mathbf{w}_j$. Assume each parameter $w_{i,k}$ (for $k = 1, \ldots, d$) is a random variable with mean $\mu_k$ and variance $\sigma_k^2$. Due to CLT, for large $d$ (typical in ML models), the difference $\mathbf{w}_i - \mathbf{w}_g$ approximates a multivariate normal distribution $\mathcal{N}(0, \Sigma)$, where $\Sigma$ is the covariance matrix. The squared $L_2$ norm $||\mathbf{w}_i - \mathbf{w}_g||_2^2$ follows a chi-squared distribution with $d$ degrees of freedom. For large $d$, this converges to $\mathcal{N}(d, 2d)$ by CLT. The square root ($L_2$ norm) then approximates $\mathcal{N}(\sqrt{d - 1/2}, \sqrt{1/4})$ via the delta method. For $L[i] \sim \mathcal{N}(\mu, \sigma)$, the one-sided tail probability satisfies Mill's inequality $P(L[i] > \mu + \lambda\sigma) \leq \frac{1}{\sqrt{2\pi}\lambda} e^{-\lambda^2/2}$. Let

$Z = \frac{L[i]-\mu}{\sigma} \sim \mathcal{N}(0,1)$. Then $P(Z > \lambda) = \int_{\lambda}^{\infty} \frac{1}{\sqrt{2\pi}} e^{-z^2/2} dz \leq \frac{1}{\sqrt{2\pi}\lambda} e^{-\lambda^2/2}$, where the inequality follows from the bound $\int_{\lambda}^{\infty} e^{-z^2/2} dz \leq \frac{1}{\lambda} e^{-\lambda^2/2}$. $\qquad\square$

## D  EFFECTIVENESS OF THE $3\sigma$ RULE

The effectiveness of the $3\sigma$ rule in identifying malicious models is supported both theoretically and empirically for the following reasons: *i*) When client datasets are i.i.d., the parameters of local models are known to follow a normal distribution (Baruch et al., 2019; Chen et al., 2017; Yin et al., 2018); *ii*) Even under non-i.i.d. settings, the Central Limit Theorem (CLT) (Rosenblatt, 1956) ensures that local models tend to approximate a normal distribution, especially when the number of clients is at least 30 (Chang et al., 2006; of Public Health, 2001); *iii*) Even when CLT does not hold strongly (*e.g.*, the number of clients is lower than 30), prior work (Karimireddy et al., 2020; Pillutla et al., 2022) shows that local models still exhibit certain statistical features, enabling the $3\sigma$ rule to remain effective.

Furthermore, our empirical evaluations in §5 confirm the reliability of the $3\sigma$ rule even with a small number of clients. This is because: *a*) SGD introduces noise during local training, which often causes model updates to approximate normality in practice, even for a small number of clients; and *b*) The *evilness level* of each local model aggregates high-dimensional model parameters, smoothing out individual irregularities and leading to a distribution that empirically resembles a Gaussian.

## E  IMPORTANCE LAYERS

Real-world deployments may face additional challenges due to: *i*) *Storage Constraints:* FL systems often operate in resource-constrained environments, necessitating storage-efficient solutions; *ii*) *Computational Overhead:* Integrating ZKPs for post-round validation (to be discussed in §4) introduces significant computational costs (Goldreich & Krawczyk, 1996); using full models for ZKP verification is computationally expensive and prolongs the verification time. To address these issues, we adopt segmental models, termed *importance layers* following (Fung et al., 2020), instead of using entire models as reference. An importance layer must satisfy: *i*) *Representativeness*: capturing sufficient model information with minimal size (ideally a single layer of the original model), and *ii*) *Generalizability*: applicability across diverse data distributions and model architectures. We note that the importance layer is not required to contain the maximal information compared with other layers, but should be more *informative* than the majority of the other layers. To improve the efficiency of computations, we select the second-to-last layer as the importance layer, as it retains substantial model information.

## F  EXTENSIONS TO ADAPTIVE ATTACKS

To extend REDJASPER to adaptive attacks, cached global models from previous FL rounds cannot be used, as the global model is distributed to all clients, enabling malicious participants to exploit it for attacks. To address this issue, we modify REDJASPER to operate without cached models. At the end of each round, the server estimates a global model using $m$-Krum (Blanchard et al., 2017), computed from the local models submitted in that round, and employs it as a reference for both cross-round and cross-client detection. To further improve detection accuracy, we use the full model parameters instead of the importance layer. The modified algorithms for the cross-round detection and the cross-client detection are detailed as follows.

**Cross-Round Detection.** To identify suspicious activities of each FL training round, the cross-round detection estimates a global model with $m$-Krum (Algorithm 3), where $m$ is set to half of the number of local models in the current round. Based on the adversarial assumptions of $m$-Krum and §2.1 where the majority of clients are benign, we ensure $m$-Krum to aggregate local models that are more likely to be benign. Thus, the estimated global model approximates the true global model and can serve as a reliable *golden truth*. It provides a reference for convergence; local models that deviate significantly from it are flagged as potentially disruptive. Notably, this estimated global model is used only for identifying potential attacks and does not impact the actual removal of models in the next phase, thus, it is sufficient for providing a dependable reference.

The algorithm is summarized in Algorithm 4. Upon the server receiving local models from clients, it clips the models based on a randomly selected norm, and applies $m$-Krum (Blanchard et al., 2017) to computes an estimated global model $\mathbf{w}_g^{ref}$. Then, it computes cosine similarities between each local model and $\mathbf{w}_g^{ref}$. Local models exhibiting higher similarities to these reference models are deemed more likely to be benign, while those with lower similarities are considered suspicious, activating a rigorous cross-client detection in the next phase. We note that REDJASPER just flags suspicious models without removing them, thus, REDJASPER does not rely heavily on cosine similarities.

---

**Algorithm 4** REDJASPER-Phase 1: Cross-Round Detection for Adaptive Attacks

---

**Input:** $\tau$: training round ID ($\tau \geq 0$); $\mathcal{W}^\tau$: client models of round $\tau$; $\gamma$: similarity threshold
1: **if** $\tau = 0$ **then return False**      ▷ *No previous models, activate cross-client detection by default*
2: $\mathcal{W}^\tau \leftarrow \text{clip}(\mathcal{W}^\tau)$, $\mathbf{w}_g^{\text{ref}} \leftarrow \text{Krum\_and\_m\_Krum}(\mathcal{W}^\tau, \frac{|\mathcal{W}^\tau|}{2}, \frac{|\mathcal{W}^\tau|}{2})$
3: **for all** $\mathbf{w}_i^\tau \in \mathcal{W}^\tau$ **do**
4:     $\mathcal{S}_c(\mathbf{w}_g^{\text{ref}}, \mathbf{w}_i^\tau) \leftarrow \text{get\_cosine\_similarity}(\mathbf{w}_g^{\text{ref}}, \mathbf{w}_i^\tau)$
5:     **if** $\mathcal{S}_c(\mathbf{w}_g^{\text{ref}}, \mathbf{w}_i^\tau) < \gamma$ **then return True**      ▷ *Potential attack detected*
6: **return False**      ▷ *No attack detected*

---

**Cross-Client Detection.** To extend the cross-client detection against adaptive attacks, we modify it to use the estimated global model $\mathbf{w}_g^{ref}$ from the cross-round detection as a reference instead of the cached global model from the last FL round. The modified algorithm is summarized in Algorithm 5.

---

**Algorithm 5** REDJASPER-Phase 2: Cross-Client Detection

---

**Input:** $\tau$: training round ID ($\tau = 0, 1, \ldots$); $\mathcal{W}^\tau$: local models of round $\tau$; $m$: parameter of $m$-Krum; $\lambda$: parameter of $3\sigma$ Rule; $\mathbf{w}_g^{ref}$: global reference model from the previous round.
1: $\mathbf{w}_g^{ref} \leftarrow \text{get\_global\_model\_from\_cross\_round\_check}()$
2: $\mathcal{L} \leftarrow \text{compute\_L2\_scores}(\mathcal{W}^\tau, \mathbf{w}_g^{ref})$, $\mu \leftarrow \frac{\sum_{\ell \in \mathcal{L}} \ell}{|\mathcal{L}|}$, $\sigma \leftarrow \sqrt{\frac{\sum_{\ell \in \mathcal{L}} (\ell - \mu)^2}{|\mathcal{L}| - 1}}$      ▷ *Estimate $\mathcal{N}(\mu, \sigma)$*
3: **for all** $\mathbf{w}_i \in \mathcal{W}^\tau$ **do**
4:     **if** $\mathcal{L}[i] > \mu + \lambda\sigma$ **then** Remove $\mathbf{w}_i$ from $\mathcal{W}^\tau$
5: **return** $\mathcal{W}^\tau$      ▷ *Cache and return the filtered set*

---

# G  ZKP IMPLEMENTATION

In our implementation, we use the Groth16 (Groth, 2016) zkSNARK scheme implemented in the Circom library (Contributors, 2022) for all the computations described earlier. We choose this ZKP scheme because its construction ensures constant proof size (128 bytes) and constant verification time. Because of this, Groth16 is popular for blockchain applications as it necessitates little on-chain computation. There are other ZKP schemes based on different constructions that can achieve faster prover time (Liu et al., 2021), but their proof size is bigger and verification time is not constant, which is a problem if the verifier lacks computational power, as in our case since the verifiers are the FL clients in our setting. The construction of a ZKP scheme that is efficient for both the prover and verifier is still an open research direction.

**ZKP-compatible language.** The first challenge of applying ZKP protocols is to convert the computations into a ZKP-compatible language. ZKP protocols model computations as arithmetic circuits with addition and multiplication gates over a prime field. However, our computations for our approach are over real numbers. The second challenge is that some computations such as square root are nonlinear, making it difficult to wire them as a circuit. To address these issues, we implement a class of operations that map real numbers to fixed-point numbers. To build our ZKP scheme, we use Circom library (Contributors, 2022), which compiles the description of an arithmetic circuit in a front-end language similar to C++ to the back-end ZKP protocol.

**Implementation of ZKP.** To implement ZKP in FL systems, we employ zkSNARKs (Bitansky et al., 2012), a ZKP variant with constant proof size and verification time, regardless of the size of computation. It is essential for real-world FL deployments, where clients often run under resource constraints. We note that the computations in Algorithms 1 and 2 rely heavily on linear operations,

which we translate into arithmetic circuits for ZKP compatibility. For instance, computing cosine similarity between two $n \times n$ matrices requires $\mathcal{O}(n^2)$ multiplication gates and a division operation. While division is non-trivial, we circumvent this by having the prover precompute quotients and remainders, which the circuit verifies via modular arithmetic.

**Interactivity of zkSNARKs.** In the Freivalds' algorithm (Freivalds, 1977), the prover first computes the matrix multiplication and commits to its result. Then the verifier generates and sends the random vector. This step is interactive in nature, but we can make this non-interactive using the Fiat-Shamir heuristic (Fiat & Shamir, 1986) as it is public-coin, meaning the vector is randomly selected by the verifier and made public to everyone. Therefore, the prover can instead generate this vector by setting it to the hash of matrices $A$, $B$ and $C$. With this, our entire ZKP pipeline, including the Freivalds' step can become truly non-interactive.

## H MOTIVATION FOR INTEGRATING ZKPS IN REDJASPER

ZKP enables proving to the clients that the server has correctly executed the anomaly detection process. This addresses a critical concern in FL systems, where clients cannot directly verify the server's behavior and must fully trust the server. Below, we explain the motivation for utilizing ZKP in our framework from research, industry, and system perspectives.

**Research perspective.** Existing literature has considered various adversarial models. For example, 1) clients might be malicious and submit modified models; 2) FL server might be curious about local models and want to infer sensitive information, such as original training data, or the local models; 3) clients might be curious about local models of other clients; 4) an external adversary may hack the communication channels between clients and the server and poison some client models; 5) the FL server may be hacked by external adversaries; 6) a global "sybil" may hack the whole system and control some clients by modifying their local training data, and so on.

In our paper, we assume the FL server is not fully trusted due to the complex execution environment in practical systems. There may be external adversaries or a global sybil, thus, even if the server hopes to execute the aggregation correctly, the presence of adversaries necessitates a ZKP module for verification to ensure that the server's actions are transparent and trustworthy to all clients.

**Industry perspective.** The necessity of ZKP also arises from real-world application needs. Consider, for example, FL clients that are medical institutions or hospitals holding sensitive data, such as patient medical records. These institutions may want to collaboratively train a model but be unwilling to share their raw data due to privacy concerns. Although these institutions know that the server will run an anomaly detection procedure, they may not be fully convinced that the server will honestly execute the procedure or that their models will participate in the aggregation without bias. Here, ZKP enables verification that the anomaly detection is performed correctly, even when the clients do not have access to the local models of other clients. This is critical for gaining the trust of the participating clients.

**System perspective.** Real-world FL systems with incentive mechanisms typically involve multiple components, including model aggregation, contribution assessment, anomaly detection, and more. When the FL server is not fully trusted, it becomes critical to validate these operations. In this paper, we focus specifically on anomaly detection and primarily discuss the application of ZKPs in this context. The ZKP module ensures that even if the server is not fully trusted, *e.g.*, under potential external threats, clients can have verifiable proof that the anomaly detection and potentially other components have been executed correctly, thus maintaining the integrity of the FL process.

## I EXTENSION TO CLIENT SAMPLING

Our method can work in the case of client sampling. For ease of explanation, in the main manuscript, we assume that all clients participate in aggregation in every FL round. With appropriate engineering efforts, our method can be readily extended to incorporate client selection. To handle scenarios with client selection, we can cache historical client models for the same clients across rounds, such that the server can perform cross-round detection even when clients do not participate in every round. If the cached model for a client is too old, we can use the global model from the last round as the reference model. A scenario with adversary clients that participate only once (*i.e.*, single-shot attacks)

constitutes a specific case of the client selection challenge described above. In such cases, we can use the global model from the last round as the reference model for cross-round detection.

## J Supplementary Experimental Results

The results for the importance layers of RNN, CNN, and ResNet-56 are given in Figure 11a, Figure 11b, and Figure 11c, respectively. The results for evaluations on RNN and the Shakespeare dataset is shown in Figure 11d.

**Exp 12: Evaluations against adaptive attacks under a different attack frequency.** We set the adaptive attack to occur randomly in 40 out of 100 FL rounds. We compare our approach with other defenses on the following tasks: *i*) a CV task: ResNet20 with CIFAR10; *ii*) CV task: ResNet50 with CIFAR100; and *iii*) an NLP task: RNN with the Shakespeare dataset; See Figure 12a, Figure 12b, and Figure 12c. The results show that the performance of our approach maintains high performance even when attacks occur randomly, indicating the effectiveness of our method in accurately identifying and removing malicious local models under varying attack frequencies.

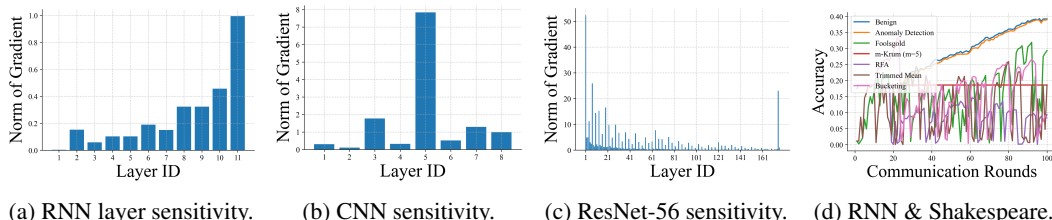

(a) RNN layer sensitivity. (b) CNN sensitivity. (c) ResNet-56 sensitivity. (d) RNN & Shakespeare.

Figure 11: Supplementary experimental results.

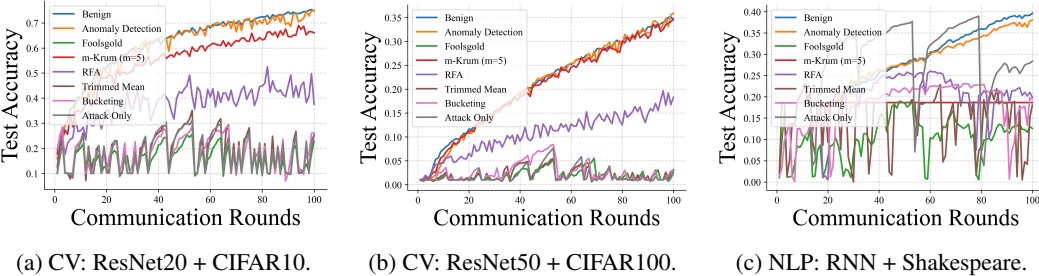

(a) CV: ResNet20 + CIFAR10. (b) CV: ResNet50 + CIFAR100. (c) NLP: RNN + Shakespeare.

Figure 12: Evaluations under a different attack frequency across different tasks.

Table 2: Cost of ZKP of different models.

| Model | Stage 1 Circuit Size | Stage 2 Circuit Size | Proving Time (s) | Verification Time (ms) |
|---|---|---|---|---|
| CNN | 476,160 | 795,941 | 33 (12 + 21) | 3 |
| RNN | 1,382,400 | 2,306,341 | 96 (34 + 62) | 3 |
| ResNet-56 | 1,536,000 | 2,562,340 | 100 (37 + 63) | 3 |

Bracketed times denote duration for cross-round detection and cross-client detection.

**Exp 13: Evaluations of ZKP verification.** We implement a prover's module which contains JavaScript code to generate witness for the ZKP, as well as to perform fixed-point quantization. Specifically, we only pull out parameters of the importance layer to represent the whole model to reduce complexity. We report the results in Table 2. The results show that the proving is efficient as we utilize importance layers, instead of entire models, for computation.

**Exp 14: Evaluations in a real-world setting.** To validate the utility and scalability of our approach in real-world applications, we utilize 20 real-world edge devices to demonstrate how our anomaly detection mechanism performs under practical constraints and settings.

In each FL round, we designate 5 devices as malicious. The FL client package is integrated into the edge nodes to fetch data from our back-end periodically. Due to the challenges posed by real-world settings, such as devices equipped solely with CPUs (lacking GPUs), potential connectivity issues, network latency, and limited storage on edge devices, we select a simple task, *i.e.*, using the MNIST dataset for a logistic regression task, to run FL training for 10 rounds, and use our proposed anomaly detection method to prevent against the random weight Byzantine attack. We also included a benign case and an attack-only case for comparison, and and the results are shown in Figure 13, with a total training time of 221 seconds. Results show that despite the presence of malicious clients and the limitations of edge devices, our approach (shown in green in Figure 13) successfully identifies and mitigates the impact of malicious local models.

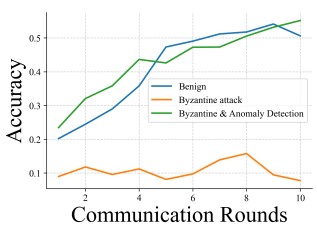

Figure 13: Evaluations on real-world edge devices.

## K  USAGE OF LLMS

This paper leverages LLMs to enhance the quality of writing, especially polishing sentences, correcting grammatical errors, and improving overall readability.

