# OpenReview forum: "Kick Bad Guys Out! Conditionally Activated Anomaly Detection in Federated Learning with Zero-Knowledge Proof Verification"
_ICLR.cc/2026/Conference — ICLR 2026 Conference Withdrawn Submission_

### Official Review · Reviewer_91Ag · 2025-10-27

**Soundness:** 2
**Presentation:** 2
**Contribution:** 2
**Rating:** 4
**Confidence:** 3

**Summary:**

This paper proposes RedJasper, a two-stage anomaly detection method for Federated Learning that conditionally activates rigorous scrutiny of suspicious local models using the $3\sigma$ rule and removes the malicious ones. The proposed method also involves zero knowledge proof verification to verify the server's detection process.

**Strengths:**

1. The paper proposes a practical design without requiring prior knowledge of attacks or attackers.

2. The proposed method integrates verification on the server's detection process via ZKPs, enhancing transparency and trust.

**Weaknesses:**

1. The integration of ZKPs introduces significant additional overhead, which limits the applicability of the proposed method.

2. The technical contribution is limited since the detection methods have been well-established by existing work. The integration of ZKPs is also not novel.

3. The number of clients is small in the experiments. The evaluation is also limited to two datasets.

**Questions:**

Please refer to the weakness part.

---

### Official Review · Reviewer_F5Rs · 2025-10-29

**Soundness:** 2
**Presentation:** 3
**Contribution:** 1
**Rating:** 4
**Confidence:** 4

**Summary:**

This paper proposes RedJasper, a two-stage anomaly detection defense for FL that addresses the practical limitations of existing defenses (e.g., interference with benign training and lack of verifiability). The first stage performs a lightweight, cross-round similarity check to detect suspicious activity. If an anomaly is suspected, the second stage is activated to identify and remove malicious models. A key feature of this method is the integration of Zero-Knowledge Proofs (ZKP) to allow clients to verify the integrity of the server's detection process. Extensive experiments demonstrate the effectiveness of RedJasper.

**Strengths:**

1.The use of ZKP to verify the server-side defense process is an innovative and important contribution, which strives to address the client-server trust problem.

2.This paper is well written and has an intuitive comparison between existing methods.

**Weaknesses:**

1.The novelty of the work is insufficient since using anomaly detection to identify malicious models is a very common practice.

2.The first stage's reliance on cosine similarity appears potentially vulnerable to advanced adaptive attacks (e.g., A3FL [1], Chameleon [2]).  Attackers could craft models to maintain high similarity, bypassing the initial detection and rendering the entire two-stage defense ineffective.  The paper's discussion of this risk is not fully convincing.

3.The evaluation fails to compare RedJasper against more recent and powerful SOTA defenses (like FLAME [3] or DeepSight [4]) or more advanced SOTA attacks.  This makes it difficult to assess its true effectiveness and relative advantage in the current security landscape.

4.The practical utility of the ZKP component is questionable due to high overhead.  The reported proving times are significant for fast FL rounds, and the paper lacks sufficient discussion on how this scales with more complex models or larger numbers of clients.

Reference:

[1] Zhang, Hangfan, et al. A3FL: adversarially adaptive backdoor attacks to federated learning. NeurIPS’ 2023.

[2] Yanbo Dai, Songze Li. Chameleon: Adapting to Peer Images for Planting Durable Backdoors in Federated Learning. ICML2023.

[3] Nguyen et al. FLAME: Taming backdoors in federated learning. USENIX Security’ 2022.

[4] Phillip Rieger, Thien Duc Nguyen, et al. DeepSight: Mitigating Backdoor Attacks in Federated Learning Through Deep Model Inspection. NDSS 2022.

**Questions:**

In the introduction, this paper mentions many existing defense methods. Why not compare the defense performance with them in the experiment?

---

### Official Review · Reviewer_jGj2 · 2025-11-01

**Soundness:** 2
**Presentation:** 2
**Contribution:** 2
**Rating:** 2
**Confidence:** 5

**Summary:**

This paper proposes REDJASPER, a novel two-stage defense mechanism for Federated Learning (FL) designed to be practical for real-world deployment. The authors argue that existing defenses suffer from critical flaws: they either rely on unrealistic assumptions (like knowing the number of attackers) or are "always on," which can degrade model accuracy even in benign, attack-free scenarios.

REDJASPER addresses this with its "conditional activation" architecture. The first stage is a lightweight "gatekeeper" that performs a cross-round check, comparing the cosine similarity of incoming models against benign references. If no suspicious activity is detected, training proceeds normally. However, if this first stage flags a potential threat, it conditionally activates the more rigorous second stage. This cross-client detection then computes an "evilness level" (L2 distance) for each suspicious model and applies the $3\sigma$ rule to statistically identify and filter out malicious outliers before aggregation.

Furthermore, to address the trust gap where clients cannot be sure the server is executing the defense honestly, REDJASPER integrates Zero-Knowledge Proofs (ZKPs). This allows clients to cryptographically verify that the server has faithfully performed the detection and filtering steps, ensuring system integrity without relying on the server's goodwill.

Experimental results show that REDJASPER effectively identifies attacks while maintaining model performance comparable to benign, attack-free scenarios.

**Strengths:**

- It correctly identifies a key practical flaw in many defenses: being "always on" (perpetually active) is suboptimal, as it can degrade model performance and in benign, attack-free scenarios. The conditional activation architecture is an elegant solution to this problem. The two-stage design is logical and clearly illustrated in Figure 1, making the proposed architecture easy to understand. Figures 1 and 3 collectively show how readily deployable the solution is, which is impressive.
- The defense is well-motivated by three fair and justified principles for a practical system: on-demand activation, accurate filtering, and verifiability [Lines: 69-74]. Table 1 effectively summarizes how REDJASPER compares to SOTA defenses on these practical features.
- The integration of Zero-Knowledge Proofs (ZKPs) is a significant and relatively uncommon contribution in this area. It addresses the critical issue of server trust by providing clients with a cryptographic method to ensure the defense is being executed faithfully.
- The experimental evaluation is thorough. It covers a large range of models and datasets, including diverse tasks in both Computer Vision (FEMNIST, CIFAR-10, CIFAR-100) and NLP (Shakespeare), which supports the generalizability of the approach.
- The results look strong across the board (e.g., Figures 7, 8, 9). They consistently show REDJASPER maintaining high accuracy, very close to the benign (no attack) baseline, while many competing defenses like m-Krum, RFA, and Trimmed Mean collapse under the same attacks.

**Weaknesses:**

- The paper emphasizes how previous defenses are based on unrealistic assumptions about the number of malicious clients. However, this defense just swaps one assumption for another. Getting rid of the assumption on the number of malicious clients makes it necessary for the protocol to rely on several thresholds, namely the $3\sigma$  rule and the cosine similarity threshold $\gamma$. These thresholds seem harder and less intuitive to set than a single assumption on the maximum number of malicious clients.
- In the cross-round detection, the defense relies on cosine similarity between two models (their weights), which is not a robust metric in FL. Comparing two gradients via cosine similarity can reveal differences in direction (e.g., gradient descent vs. ascent), but comparing two models effectively sets the reference as the origin. This means a benign and a malicious model could be indistinguishable if they are in a similar direction, even if their updates are contradictory.
- For such a threshold-based defense, an evaluation against smart, adaptive whitebox attacks is necessary. An adversary could theoretically develop an attack that stays just within the thresholds ($3\sigma$), allowing it to keep steering the model toward a malicious target over time. Such an evaluation was lacking.
- The model poisoning attacks chosen for evaluation - random weight and zero weight, are very weak and outdated. The defense was not tested against more advanced, gradient ascent attacks [1,2], which are more relevant baselines.
- The primary experiments are done with only 10 clients, which is substandard and too low to draw general conclusions. This small scale is particularly problematic for studying the number of attackers; for instance, in the backdoor attack experiment, a 10% malicious ratio means only one node was malicious, which constitutes a very weak attack.
- While the results in Figures 7 and 8 show success, the work's deployable nature and good implementation make it a good system for FL against weak, general attackers. However, it is not a convincing solution against smart or adaptive attackers.

[1]: Shejwalkar, Virat, and Amir Houmansadr. "Manipulating the byzantine: Optimizing model poisoning attacks and defenses for federated learning." NDSS. 2021.
[2]: Shejwalkar, Virat, et al. "Back to the drawing board: A critical evaluation of poisoning attacks on production federated learning." 2022 IEEE symposium on security and privacy (SP). IEEE, 2022.

**Questions:**

- In Section 3.2, the global model from the previous round is used as a reference. In the unfortunate case that an attack succeeded in the last round, this reference would be poisoned. What guardrails can be set to prevent this poisoned reference from supporting the attacker in the current round?
- The paper states the evaluation used 8 A100 GPUs, which seems like excessive hardware for simulating a primary experiment of only 10 clients. Why was this level of computational power required?

---

### Official Review · Reviewer_qxkG · 2025-11-02

**Soundness:** 2
**Presentation:** 2
**Contribution:** 2
**Rating:** 2
**Confidence:** 4

**Summary:**

The paper introduces RedJasper, a defense against poisoning attacks in federated learning relying on a two-stage approach: First a cross-round detection check to evaluate the likelihood of potential attacks, and second, a cross-client detection check to evaluate with more detail which participants seem to be malicious. A Zero Knowledge Proof (ZKP) component is also included, so that clients can cryptographically verify the server’s execution of the defense without revealing private data. The experimental evaluation includes two Byzantine and two backdoor attacks, comparing RedJasper’s performance against five baseline defenses, showing the ability of RedJasper to mitigate such attacks.

**Strengths:**

+ The approach to mitigate the attacks in two stages, one lightweight check to flag possible attacks, before going into a deeper analysis is interesting to reduce the computational burden in federated learning training, especially as in practice attacks may not occur very often.
+ The paper is well-organized and structured, with formal descriptions for the algorithms and an attempt to provide a theoretical bound on false positives.
+ Compared to other defenses against poisoning attacks in federated learning, the paper emphasizes efficiency and practical deployment, addressing the cost of complex aggregation defenses that are always on.
+ The experiments include a comprehensive set of model architectures and datasets.

**Weaknesses:**

+ Some of the assumptions are not realistic in practical federated learning settings. Thus, theorem 3.2 and the use of the 3-sigma rule rely entirely on the assumption that the “evilness scores” follow a Gaussian distribution, which does not hold in FL with non-IID local datasets or in adversarial settings: client updates can be multi-modal, heavy-tailed, and highly non-IID, especially in the presence of attackers. Consequently, the theoretical result has limited practical relevance and provides little assurance about real-world behavior. There is no empirical evidence provided by the authors to support this.
+ RedJasper relies on m-Krum to “prevent the influence of any potentially malicious models”. However, Krum is known to fail under many realistic and sophisticated attacks and has problems with non-IID data, high-dimension, and stealthy or colluding attackers. On the other hand, restricting the detection to the penultimate importance layer makes the system completely vulnerable to attacks on early layers. This is a critical vulnerability that adaptive attackers could exploit easily.
+ The ZKP component is not well motivated in the threat model. Since the server preforms the defense, clients verifying the server execution adds little value unless the server is untrusted, which is a point that the paper never assumes or discusses. This component feels quite disconnected from the rest of the defense. On the other hand, no experiments evaluate the overhead of the ZKP component, its scalability or its actual contribution to the security of the federated learning system.
+ The five baselines (m-Krum, FoolsGold, RFA, Bucketing, Trimmed Mean) used for the experiments are weak and outdated: none are recent or specifically designed to defend against backdoors. For example, some baselines that could be included are: FLAME, FLTrust, DeepSight, FLGuard. On the other hand, in the experiments, the choide of m=5 for m-Krum is arbitrary and not properly justified (possibly is not the best value for most cases in the experiments), making comparisons potentially unfair.
+ Throughout the paper there are several overstated claims. For instance, Table 1 presents RedJasper as the only method with “robust performance in no-attack scenarios” and “attack presence detection” (together with DeepSight). These statements in Table 1 are not well substantiated and arguably false. For instance, there are a number of methods, apart from DeepSight, that tackle attack presence detection in one or another way, such as FLTrust, Flame, FoolsGold, or Adaptive Federated Averaging. Similarly, there are some claims that are exaggerated and inaccurate, as for example claiming that RedJasper is “the first method that
explicitly bridges the gap between academic research and real-world applications of FL security”.
+ The four attacks selected are basic baselines and do to provide a strong evaluation of the robustness of modern defenses. On the other hand, the adaptive attack used in the paper is not defined. Thus, it is difficult to interpret the results and assess their validity.
+ Following the previous comment, the experimental evaluation does not provide analyses varying the non-IID level of the local datasets.
+ Although the paper claims that the server is not fully trusted, the authors do not specify what misbehavior is being protected against or why ZPK verification is needed. Thus, since the defense logic still runs on the server, the ZPK integration feels ad hoc. Adding complexity without a clearly defined adversarial scenario. I believe this part can be better motivated.

**Questions:**

+ How exactly are the adaptive attacks implemented in the paper?
+ What is the concrete threat model for which ZKP verification is necessary? In other words, under what assumptions can the server be dishonest?
+ What would happen if the attacker targets unmonitored layers?
+ Can you empirically demonstrate that “evilness scores” follow an approximately Gaussian distribution in a practical scenario with non-IID settings and the presence of sophisticated adversaries?

---

### Note · Authors · 2025-12-30

I have read and agree with the venue's withdrawal policy on behalf of myself and my co-authors.